# Sensitivity to Immune Checkpoint Blockade in Advanced Non-Small Cell Lung Cancer Patients with *EGFR* Exon 20 Insertion Mutations

**DOI:** 10.3390/genes12050679

**Published:** 2021-04-30

**Authors:** Giulio Metro, Sara Baglivo, Guido Bellezza, Martina Mandarano, Alessio Gili, Giovanni Marchetti, Marco Toraldo, Carmen Molica, Maria Sole Reda, Francesca Romana Tofanetti, Annamaria Siggillino, Enrico Prosperi, Antonella Giglietti, Bruna Di Girolamo, Miriam Garaffa, Francesca Marasciulo, Vincenzo Minotti, Marco Gunnellini, Annalisa Guida, Monica Sassi, Angelo Sidoni, Fausto Roila, Vienna Ludovini

**Affiliations:** 1Medical Oncology, Santa Maria della Misericordia Hospital, via Dottori 1, 06156 Perugia, Italy; sara.baglivo@ospedale.perugia.it (S.B.); carmen.molica@ospedale.perugia.it (C.M.); mariasole.reda@ospedale.perugia.it (M.S.R.); francesca.tofanetti@ospedale.perugia.it (F.R.T.); annamaria.siggillino@ospedale.perugia.it (A.S.); miriam.garaffa@libero.it (M.G.); marasciulofrancesca@libero.it (F.M.); vincenzo.minotti@ospedale.perugia.it (V.M.); vienna.ludovini@ospedale.perugia.it (V.L.); 2Section of Anatomic Pathology and Histology, Department of Medicine and Surgery, University of Perugia, Piazza Lucio Severi 1, 06132 Perugia, Italy; guido.bellezza@unipg.it (G.B.); martina.mandarano@unipg.it (M.M.); angelo.sidoni@unipg.it (A.S.); 3Public Health Section, Department of Experimental Medicine, University of Perugia, Piazza Lucio Severi 1, 06132 Perugia, Italy; alessio.gili@gmail.com; 4Division of Pathology, Santa Maria Hospital, via Tristano di Joannuccio 1, 05100 Terni, Italy; g.marchetti@aospterni.it; 5Division of Pathology, San Giovanni Battista Hospital, via Massimo Arcamone, 06034 Foligno, Italy; marco.toraldo@uslumbria2.it; 6Division of Pathology, Città di Castello Hospital, Via Luigi Angelini 10, 06012 Città di Castello, Italy; enrico.prosperi@uslumbria1.it; 7Hematology and Oncology Unit, San Giovanni Battista Hospital, via Massimo Arcamone, 06034 Foligno, Italy; antonella.giglietti@uslumbria2.it (A.G.); monica.sassi@uslumbria2.it (M.S.); 8Oncologic Day Hospital, Santa Maria della Stella Hospital, Località Ciconia, 05018 Orvieto, Italy; bruna.digirolamo@uslumbria2.it; 9Medical Oncology, Gubbio and Gualdo Tadino Hospital, Largo Unità d’Italia, 06024 Branca, Italy; marco.gunnellini@uslumbria1.it; 10Medical Oncology, Santa Maria Hospital, via Tristano di Joannuccio 1, 05100 Terni, Italy; a.guida@aospterni.it; 11Medical Oncology, Santa Maria della Misericordia Hospital, University of Perugia, Piazza Lucio Severi 1, 06132 Perugia, Italy; fausto.roila@unipg.it

**Keywords:** *EGFR* exon 20 insertion mutations (Ex20ins), immune checkpoint blockade (ICB), immunotherapy, non-small-cell lung cancer (NSCLC), PD-L1

## Abstract

Besides platinum-based chemotherapy, no established treatment option exists for advanced non-small-cell lung cancer (NSCLC) patients with *EGFR* exon 20 (Ex20ins) insertion mutations. We sought to determine the clinical outcome of patients with this *EGFR* mutation subtype in the immunotherapy era. Thirty NSCLCs with *EGFR* Ex20ins mutations were identified, of whom 15 had received immune checkpoint blockade (ICB) treatment as monotherapy (N = 12), in combination with chemotherapy (N = 2) or with another immunotherapeutic agent (N = 1). The response rate was observed in 1 out of 15 patients (6.7%), median progression-free survival (PFS) was 2.0 months and median overall survival (OS) was 5.3 months. A trend towards an inferior outcome in terms of PFS and OS was observed for patients receiving ICB treatment in the first versus second line setting (PFS: 1.6 months versus 2.7 months, respectively, *p* = 0.16—OS: 2.0 months versus 8.1 months, respectively, *p* = 0.09). Median OS from the time of diagnosis of advanced disease was shorter for patients treated with ICB versus those who did not receive immunotherapy (12.9 months versus 25.2 months, respectively, *p* = 0.08), which difference remained associated with a worse survival outcome at multivariate analysis (*p* = 0.04). Treatment with ICB is poorly effective in NSCLCs with *EGFR* Ex20ins mutations, especially when given in the first-line setting. This information is crucial in order to select the optimal treatment strategy for patients with this subtype of *EGFR* mutation.

## 1. Introduction

Two major treatment paradigms have been recently established for the management of advanced non-small-cell lung cancer (NSCLC), either targeted therapies based on the inhibition of aberrant oncogenic drivers or immunotherapy based on the blocking of immunosuppressive checkpoints [1]. Targeting the *epidermal growth factor receptor* (*EGFR*) mutation with an EGFR-tyrosine kinase inhibitor (-TKI) is a highly effective treatment strategy for patients with certain types of *EGFR* mutation, such as those located at exon 19 (deletions) or 21 (L858R point mutations), which comprise approximately 90% of all *EGFR* mutations [2]. Similarly, some uncommon *EGFR* mutations may still respond to treatment with an EGFR-TKI (i.e., G719, L861Q and S861Q), although to a less extent compared with common *EGFR* exon 19 deletion/L858R point mutations. By contrast, NSCLCs with *EGFR* exon 20 insertion mutations (Ex20ins), which constitute a heterogeneous group of genetic in-frame insertions in exon 20 of *EGFR*, are generally insensitive (with very few exceptions) to EGFR-TKI targeted treatment [3]. Therefore, platinum-based chemotherapy is to be considered the standard first-line treatment for NSCLCs with *EGFR* Ex20ins mutations; however, this results in a dismal prognosis as no other validated options are available at the time of disease progression.

Treatment with immune checkpoint blockade (ICB) with programmed cell death-1 (PD-1)/programmed cell death ligand-1 (PD-L1)/cytotoxic T-lymphocyte antigen 4 (CTLA-4) monoclonal antibodies has led to unprecedented durable clinical benefit for NSCLC, but response rates are low for patients with oncogenic drivers such as *EGFR* mutations [4,5]. However, the impact of uncommon subtypes of *EGFR* mutations on the efficacy of treatment with ICB has been less explored. While uncommon non-Ex20ins *EGFR* mutations seem to experience some benefit from ICB treatment, little is known about the correlation existing between NSCLCs with *EGFR* Ex20ins mutations and response to immunotherapy [6]. In more detail, it is not known whether NSCLCs harboring *EGFR* Ex20ins mutations are suitable for treatment with ICB either in the initial ‘naïve’ setting or resistant clinical scenario after platinum-based chemotherapy.

In this multicenter study we aimed to evaluate the clinicopathological and molecular characteristics of advanced NSCLC patients with *EGFR* Ex20ins in order to associate them with clinical outcome according to ICB treatment. The final intent of this analysis was to assess whether immunotherapy could represent a valuable therapeutic strategy for this group of patients.

## 2. Materials and Methods

### 2.1. Study Population

Patients with advanced (either locally advanced inoperable or metastatic) NSCLC and *EGFR* Ex20ins mutations who had received at least one anti-cancer therapy were identified from a prospectively maintained institutional database. Patients were enrolled at 5 different Institutions (Santa Maria della Misericordia Hospital in Perugia, San Giovanni Battista Hospital in Foligno, Santa Maria Hospital in Terni, Santa Maria della Stella Hospital in Orvieto and Gubbio and Gualdo Tadino Hospital in Branca). The medical records of these patients were retrospectively collected, and only patients with clinically available data were considered to be eligible. All patients whose *EGFR* Ex20ins mutation were initially detected by direct sequencing were retrospectively subjected to confirmation by next generation sequencing (NGS) (see below), which was implemented at Perugia Hospital since June 2016.

### 2.2. NGS Analysis

NGS analysis was performed in the Molecular Biology Laboratory of Medical Oncology Division at S. Maria della Misericordia Hospital in Perugia, Italy. For DNA extraction, 2–5 sections of 10 µm-thick formalin-fixed paraffin-embedded (FFPE) tissues were prepared. One slide at the beginning of each serial section was stained with hematoxylin-eosin and histopathologically examined to determine the tumor cell content. Only samples with a sufficient tumor cell content were included in the study. After macrodissection of the tumor area, DNA was isolated using a QIAamp DNA FFPE Tissue kit (Qiagen GmbH, Hilden, Germany) on QiaCube robotic workstation according to the manufacturer’s instructions. DNA concentrations were determined with the Qubit DNA HS Assay Kit and Qubit 3.0 Fluorometer (Thermo Fisher Scientific, Carlsbad, CA, USA). Starting from 10 ng of DNA, libraries were prepared manually using the Ion AmpliSeq™ Library kit 2.0 and Ion AmpliSeq™ Colon and Lung Cancer Research Panel v2 (Thermo Fisher Scientific, Carlsbad, CA, USA) to amplify hotspots and targeted and regions of 22 genes (*KRAS*, *EGFR*, *BRAF*, *PIK3CA*, *AKT1*, *ERBB2*, *PTEN*, *NRAS*, *STK11*, *MAP2K1*, *ALK*, *DDR2*, *CTNNB1*, *MET*, *TP53*, *SMAD4*, *FBX7*, *FGFR3*, *NOTCH1*, *ERBB4*, *FGFR1* and *FGFR2*). Equalized purified libraries were combined and diluted to obtain a 16-Ion Express Barcode 30pM single pool. Template preparation and enrichment were performed using the Ion Chef™ system and Ion 510™, Ion 520™ and Ion 530™ Kit—Chef (Thermo Fisher Scientific, Carlsbad, CA, USA). Sequencing was performed on S5 System using an Ion 510 Chip and Ion S5™ sequencing kit. Signal processing and base calling ware carried out basing analysis on the default base-caller parameters of Torrent Suite (v.5.12). Variants with a quality < 30 were filtered out. NGS data analysis was performed using Ion Reporter. The limit of detection (LOD) for single nucleotide variants (SNVs), insertions/deletions and splice site mutations was ≥3% mutant allele frequency (MAF) with a minimum depth of 500×. The frequency of each mutant allele was recorded. Amplicon reads were reviewed with an Integrative Genomics Viewer (IGV) allowing fir visual inspection of the coverage of the interested regions. Alignment and variant calling were performed using human reference genome 19 (hg19).

### 2.3. Tumor PD-L1 Analysis

Tumor PD-L1 analysis was performed in the Section of Anatomic Pathology and Histology of Perugia University, Italy. Immunohistochemistry (IHC) for PD-L1 was performed using the PD-L1 22C3 pharmDx kit (Dako North America Inc., Carpinteria, CA, USA) on the Dako Autostainer Link 48, according to the manufacturers’ instructions. Unstained tissue section 4-μm thick were prepared from the most representative area of each case. At least 100 viable tumor cells were required for a valid interpretation of PD-L1 staining. Slides were counterstained with Mayer’s hematoxylin. Results were evaluated with known positive and negative tissue controls. The percentage of PD-L1 expression on invasive tumor cells was calculated as the number of viable invasive carcinoma cells showing membranous staining of any intensity divided by the total number of viable invasive carcinoma cells.

### 2.4. Statistical Analysis

The sample size was estimated according to the expected enrollment of the participating centers and to the purely descriptive intent of the study itself. The primary endpoints were to assess the clinicopathological and molecular characteristics of advanced NSCLC patients with *EGFR* Ex20ins mutations and describe their clinical outcome according to ICB treatment in terms of response rate (RR), progression-free survival (PFS) and overall survival (OS). The secondary endpoints were to evaluate the overall clinical outcome of patients with *EGFR* Ex20ins mutations according to whether they had received ICB treatment, as well as to perform univariate and multivariate analyses for OS in all patients in order to assess whether any baseline clinical factors would affect clinical outcome. Responses were assessed with RECIST (version 1.1) criteria [7]. Progression-free survival was the time elapsing between the start of ICB treatment and disease progression at any site or death in the absence of documented disease progression, while OS was calculated from the start of ICB treatment (for the primary endpoint) or from the diagnosis of advanced disease (for the secondary endpoints) until death for any cause. Descriptive statistics were calculated including frequencies, percentages, frequency tables for categorical variables, median and means ± standard deviation for quantitative variables. Categorical variables were evaluated by Chi-square or Fisher’s exact test when appropriate. The Kaplan–Meier method was used to analyze OS, PFS and to estimate medians of survival with two-sided 95% confidence intervals (CI). Survival curves were compared using the log-rank test. Cox regression model (univariate and multivariate) was applied to estimate the Hazard Ratio (HR) and 95% CI and to identify prognostic factors independently associated with survival times. Stepwise backward-selection was used for elimination of variables from the regression model to find a reduced model that best explains the data. All the covariates found to be statistically significant in the univariate approach were included in the multivariate mode. A *p*-value of less than 0.05 was considered to be statistically significant. Statistical analyses were performed with STATA v. 16.1 (StataCorp LP, College Station, TX, USA). The procedures followed were in accordance with rules of the local Ethics Committee and the Declaration of Helsinki. The study was approved by the Ethical Committee CEAS Umbria. (prot. No. 21274).

## 3. Results

### 3.1. Patients

From August 2008 until October 2020, 30 patients harboring an *EGFR* Ex20ins mutation were eligible for inclusion in the study. Appendix A shows the study flowchart. The A767_V769dup and H773_V774insNPH variants were the most frequently identified variants, comprising 34% of *EGFR* Ex20ins cases (N = 5 17% and N = 5 17%, respectively). *EGFR* Ex20ins patients had at least one concomitant mutation in 17 cases (56.7%), including *TP53* (N = 14, 46.7%) and *SMAD4* (N = 2, 6.7%). Figure 1A shows the distribution of *EGFR* Ex20in variants, while Figure 1B indicates the distribution of *EGFR* Ex20ins mutations along with concomitant genetic alterations found at NGS. Table 1 lists the patients’ characteristics and classifies them according to whether they had received immunotherapy (N = 15). In all cases the histological subtype consisted of adenocarcinoma. Overall, median age was 59 years, slightly more than half of patients were never smokers, and approximately one third of patients had a PD-L1 ≥ 1%. Apparently, there were no statistically significant differences between patients with *EGFR* Ex20ins mutations who received ICB treatment versus those who did not, except for the presence of a higher number of patients with PD-L1 ≥ 1% in the immunotherapy group. Further details on the type of treatments received besides immunotherapy for each group of patients are described in Appendix A. Table 2 shows some other key characteristics of the 15 patients who were treated with ICB treatment. Eighty percent of patients (12/15) were treated with an anti-PD1 or anti-PD-L1 agent as monotherapy, while immunotherapy was administered as first line treatment in 40% of patients (6/15). Of them, 4 patients received pembrolizumab as monotherapy because of PD-L1 ≥ 50%, while 2 patients who had a PD-L1 of 25% and <1% were treated with platinum-based chemotherapy in combination with pembrolizumab.

### 3.2. Activity of Treatment with Immune Checkpoint Blockade

Table 3 shows the activity of treatment with the immune checkpoint blockade. Only one patient (6.7%) experienced a partial response, with 11 patients (73.3%) showing progressive disease as their best response. Figure 1C shows the distribution of response according to the type of *EGFR* Ex20ins variant. The only patient who responded to ICB treatment was found to have a V769_D770insAV. However, the same mutation was found in a patient who experienced PD as best response. Of the 6 patients who received treatment with immune ICB as first line therapy, 4 patients were evaluable for response. Of them there were 3 patients treated with pembrolizumab as monotherapy and 1 patient treated with platinum-based chemotherapy in combination with pembrolizumab, all of whom underwent a progressive disease. The other 2 patients treated in first line with ICB (1 with pembrolizumab as monotherapy and 1 with platinum-based chemotherapy in combination with pembrolizumab) were not evaluable for response because of a rapid deterioration of clinical conditions after the administration of only one cycle of treatment, with subsequent death in the absence of radiological disease reassessment. Only 2 of the 6 patients treated in the first line setting (both of whom were treated with pembrolizumab as monotherapy) were able to receive a second line treatment consisting of platinum-based chemotherapy in one case and the EGFR-TKI inhibitor poziotinib in another case. At a median follow-up of 4.6 months (range 0.7–21.6), median PFS was 2.0 months (95% CI 0.6–2.7), while median OS was 5.3 months (95% CI 1.8–12.5) (Figure 2A,D). Patients treated in the first line setting experienced a worse outcome in terms of both PFS and OS; however, they did not reach statistical significance likely because of the low number of patients being compared (PFS: 1.6 months for first line versus 2.7 months for second line, *p* = 0.16—OS: 2.0 months for first line versus 8.1 months for second line, *p* = 0.09) (Figure 2B,E). By contrast, no apparent difference was noted for PFS and OS by PD-L1 status (PFS: 2.7 months for PD-L1 < 1% versus 2.0 months for PD-L1 ≥ 1%, *p* = 0.39—OS: 12.5 months for PD-L1 < 1% versus 4.2 months for PD-L1 ≥ 1%, *p* = 0.58) (Figure 2C,F).

### 3.3. Overall Survival of All Patients and by Treatment with ICB

We went on to evaluate the OS from the time of diagnosis of advanced disease for all patients with *EGFR* Ex20ins mutations and according to the type of treatment received. At a median follow-up of 13.2 months (range 1.4–40.4), the median OS for all patients was 17.2 months (Figure 3A). When patients were classified by whether they had received treatment with ICB, median OS was shorter, at the limit of significance, for patients treated with immunotherapy versus those who did not receive ICB treatment, being 12.9 months (95% CI 13.3–35.0) versus 25.2 months (95% CI 3.2–28.2) (*p* = 0.08), respectively (Figure 3B). At multivariate analysis for OS, ICB treatment remained significantly associated with shorter OS (*p* = 0.04), along with other parameters such as performance status of 2 and having received a number of treatment lines ≤ 2 (Table 4).

## 4. Discussion

In this multicenter study we evaluated the effects of ICB treatment on the outcome of advanced NSCLC patients with *EGFR* Ex20ins mutations and found very poor clinical results of immunotherapy. Response rate was 6.7% (1/15), while PD was observed in 73.3% (11/15) of patients (Table 3). Additionally, median PFS was 2.0 months, and median OS was 5.3 months (Figure 2A,D). These very modest results in terms of clinical outcomes suggest that ICB treatment is poorly effective in patients with *EGFR* Ex20ins mutations. Data available in the literature for NSCLCs with *EGFR* Ex20ins mutations treated with ICB are very scant and often refer to a limited number of patients. Chen et al. found a RR of 22.2% (2/9) in patients treated with anti-PD-1/PD-L1 monoclonal antibodies [8]. In another report, Lau et al. observed a RR of 50.0% (3/6) and a median PFS of 4.8 months in patients undergoing ICB treatment [9]. Although these two works seem to suggest some benefit from treatment with immune checkpoint inhibitors in NSCLCs with *EGFR* Ex20ins mutations, it is authors’ opinion that they are to be considered less reliable as compared with our results that have been obtained on a larger number of patients (N = 15). By contrast, a study by Hastings et al. which took into account 28 patients with *EGFR* Ex20ins mutations treated with ICB reported a clinical outcome very similar to our study, with a RR of 10.7% (3/28), a PD of 64.3% (18/28), a median PFS of 1.9 months and a median OS of 5.5 months [10]. In addition, the same authors also associated the tumor mutation burden (TMB), which is an established marker of sensitivity to ICB, with a given *EGFR* mutation subtype, and found that *EGFR* Ex20ins mutations bear the lowest median levels of TMB (2.8 mutations/Mb in 19 evaluated patients) when compared with all other *EGFR* mutation subtypes, including common *EGFR* exon 19 deletion/L858R point mutations [10]. Accordingly, another study found low median levels of TMB (3.6 mutations/Mb) in 260 patients with *EGFR* Ex20ins mutations, with only 4% and 0.7% of these patients harboring intermediate-high (10 to 20 mutations/Mb) and high (>20 mutations/Mb) TMB levels, respectively [11]. Certainly, the low levels TMB observed in NSCLCs with *EGFR* Ex20ins mutations are compatible with the predominant never/light smoking history reported by these patients and provide a strong molecular rationale in order to explain the poor activity of ICB treatment that we observed in this group of patients.

In the present study, we could not find any correlation between the sensitivity to treatment with ICB and a given Ex20ins variant (Figure 1C). In fact, the only patient who responded to immunotherapy was found to have the same Ex20ins variant as a patient who experienced progressive disease (V769_D770insAV), with both patients receiving an anti-PD-(L)1 as monotherapy. Similarly, the only patient who had SD on immunotherapy possessed the same variant as two other patients who experienced progressive disease (H773_V774insNPH). This suggests that factors other than the type of Ex20ins variant could explain the reason why one patient responded to ICB treatment in our series. Recent studies have suggested that NSCLCs with uncommon *EGFR* mutations, including *EGFR* Ex20ins mutations, tend to be positive for CD8+ tumor infiltrating lymphocytes in approximately 50% to 60% of cases [8,12]. However, no correlation has been so far demonstrated between the levels of CD8+ tumor infiltrating lymphocytes and sensitivity to immunotherapy in NSCLC patients with uncommon *EGFR* mutations, which could be warranted in future correlative studies.

Whether ICB is active as first-line treatment for NSCLCs with *EGFR* Ex20ins mutations is a relevant question, given the fact that no standard therapeutic option exists for these patients besides chemotherapy. Importantly, we noted an inferior PFS and OS for patients who received ICB as first line treatment, although the difference did not reach statistical significance likely due to the small number of patients that were compared (6 versus 9 patients treated in the first and second line settings, respectively) (Figure 2B,E). These data suggest the absence of activity of first-line immunotherapy especially when used as single agent for patients with PD-L1 ≥ 50% (among the four PD-L1 ≥ 50% patients treated with pembrolizumab monotherapy in the first line setting 3/3 underwent PD, while another patient underwent rapid deterioration of clinical conditions leading to death). At the present time, data on the use of single agent ICB in the first-line treatment of *EGFR*-mutant NSCLC have been published by Lisberg et al. and suggest a total lack of efficacy of pembrolizumab in the initial ‘naïve’ setting despite the presence of PD-L1 expression ≥ 1% [13]. Interestingly, the same study included 2 patients with *EGFR* Ex20ins mutations and PD-L1 ≥ 50%, both of whom experienced PD as the best response.

Importantly, in our case series PD-L1 status was shown to have neither a predictive nor a prognostic role in relation to response to ICB treatment (Figure 2C,F). This absence of predictivity for PD-L1 is in line with other data that suggest the presence of an *EGFR* mutation may abrogate the predictive value of PD-L1 expression on response to ICB treatment [10,14]. The less relevant impact of PD-L1 expression in *EGFR*-mutant NSCLCs is also supported by the fact that lower levels of PD-L1 expression are generally present in case of *EGFR* mutation as compared with *EGFR* wild type tumors, as well as in comparison with NSCLCs with other driver mutations [14,15]. This is in line with the hypothesis that positive PD-L1 expression in patients with *EGFR*-mutant NSCLC more likely represents a constitutive activation of PD-L1 signal rather than a marker of adaptive immune response.

Overall, we also found that having received ICB treatment was associated with a worse survival from the time of diagnosis of advanced disease (Figure 3A,B), which was a factor found to be prognostic in the multivariate analysis. Apparently, this finding was not due to outperformance of patients in the group not receiving immunotherapy, as similar median survivals ranging between 24 and 26 months have been reported for patients with *EGFR* Ex20ins mutations not treated with immunotherapy [16,17]. By contrast, we can speculate that the poor survival of the immunotherapy group could be due to the fact that 4 out of the 15 patients treated with immunotherapy never received platinum-based chemotherapy and that these patients were less frequently administered platinum doublets (Table 1 and Appendix A). In addition, whether immunotherapy led to hyperprogressive disease in some cases that, in turn, could have had a negatively impact on the overall survival of patients treated with ICB, is a possibility that cannot be ruled out. Thus far, hyperprogressive disease has been already described in the setting of oncogene addicted NSCLC either at a preclinical or clinical level [8,18,19].

By combining multiple patients from different institutions, we managed to include 30 patients with *EGFR* Ex20ins mutations in this study, which might be considered a relatively small sample size. Other possible limitations of this analysis are the retrospective nature and the inclusion of a population of patients heterogeneously treated with different immune checkpoint inhibitors either alone or in combination. In fact, while the majority of patients had received an anti-PD1 or anti-PD-L1 as monotherapy (N = 12), few patients had received ICB combination regimens (N = 3) either an anti-PD1 + anti-CTLA4 (N = 1) or an anti-PD1 + platinum-based chemotherapy (N = 2). Therefore, it is important that future studies will address the role of ICB given in combination regimens in NSCLCs with *EGFR* Ex20ins mutations, especially when given in combination with platinum-based chemotherapy.

## 5. Conclusions

In the present study, treatment with ICB was associated with a very poor clinical outcome in advanced NSCLCs with *EGFR* Ex20ins mutations, with only one response (6.7%) and a dismal median PFS (2.0 months) and OS (5.3 months) observed. The worst treatment outcome was reported when ICB was used in the first line setting. In addition, patients treated with immunotherapy had a significantly worse OS from the time of diagnosis of advanced disease as compared with patients who never received ICB, which suggests that immunotherapy should be the least considered option in patients with *EGFR* Ex20ins mutation, to be reserved as a treatment option only after having administered standard platinum-based chemotherapy. On the other hand, more clinical data are needed in order to support the use of ICB in combination with platinum-based chemotherapy as first line treatment for these patients.

## Figures and Tables

**Figure 1 genes-12-00679-f001:**
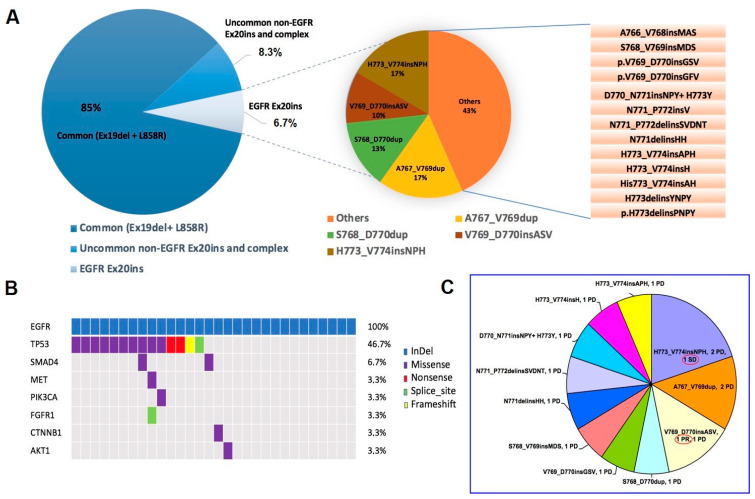
Distribution of types of EGFR mutations and EGFR Ex20ins mutations in the study cohort (N = 445) (**A**); the distribution of EGFR Ex20ins mutations and concomitant mutations stratified according to NGS sequencing and type of alteration. Each column corresponds to one of the 30 patients (**B**); the distribution of EGFR Ex20ins variants and the corresponding response to immune checkpoint blockade. Abbreviations: PR, Partial Response; SD, Stable Disease; PD, Progressive Disease (**C**).

**Figure 2 genes-12-00679-f002:**
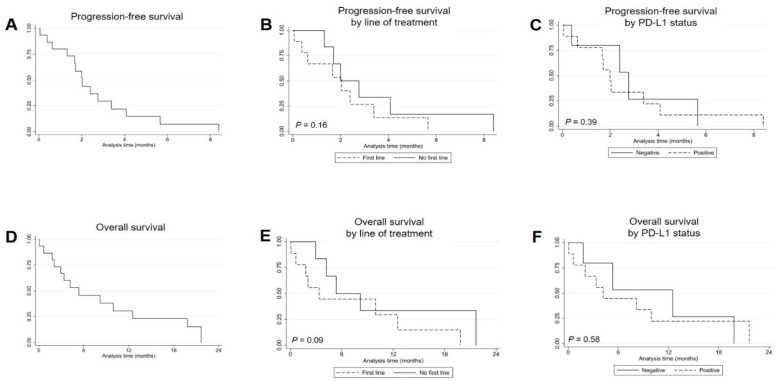
Kaplan–Meier curves of progression-free survival in all patients treated with immune checkpoint blockade (**A**) and by line of treatment (**B**) or PD-L1 status (**C**); Kaplan–Meier curves of overall survival in all patients treated with immune checkpoint blockade (**D**) and by line of treatment (**E**) or PD-L1 status (**F**).

**Figure 3 genes-12-00679-f003:**
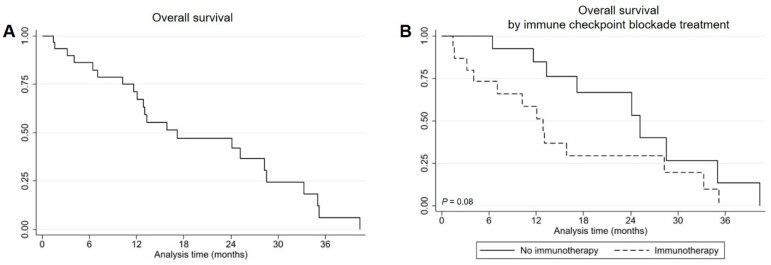
Kaplan–Meier curves of overall survival in all patients regardless of treatment with immune checkpoint blockade (**A**) and by immune checkpoint blockade treatment (**B**).

**Table 1 genes-12-00679-t001:** Characteristics of patients.

Characteristic	Total N = 30 (%)	Immunotherapy N = 15 (%)	No Immunotherapy N = 15 (%)	*p*-Value
**Median Age, Years (Range)**	59 (33–88)	58 (45–79)	64 (33–88)	0.15
**Gender**				0.71
Male	13 (43.3)	6 (40.0)	7 (46.7)	
Female	17 (56.7)	9 (60.0)	8 (53.3)	
**Smoking history**				0.71
Ever	13 (43.3)	6 (40.0)	7 (46.7)	
Never	17 (56.7)	9 (60.0)	8 (53.3)	
**Stage IV at first diagnosis**				0.28
Yes	26 (86.7)	14 (93.3)	12 (80.0)	
No	4 (13.3)	1 (6.7)	3 (20.0)	
**PS at diagnosis of advanced disease**				0.36
0–1	24 (80.0)	11 (73.3)	13 (86.7)	
2	6 (20.0)	4 (26.7)	2 (13.3)	
**Lines of treatment**				0.44
≤2	20 (66.7)	9 (60.0)	11 (73.3)	
>3	10 (33.3)	6 (40.0)	4 (26.7)	
**Any platinum-based chemotherapy treatment**				0.51
Yes	23 (76.7)	11 * (73.3)	12 (80.0)	
No	7 (23.3)	4 (26.7)	3 (20.0)	
**Any EGFR-TKI treatment**				0.26
Yes	11 (36.7)	4 (26.7)	7 (46.7)	
No	19 (63.3)	11 (73.3)	8 (53.3)	
**PD-L1 status**				**0.007**
≥1%	10 (33.3)	9 ** (60.0)	1 *** (6.7)	
<1%	15 (50.0)	5 (33.3)	10 (66.7)	
Unknown	5 (16.7)	1 (6.7)	4 (26.6)	
***TP53* mutation**				0.19
Present	14 (46.7)	8 (53.3)	6 (40.0)	
Absent	13 (43.3)	7 (46.7)	6 (40.0)	
Unknown	3 (10.0)	0	3 (20.0)	

EGFR-TKI, epidermal growth factor receptor-tyrosine kinase inhibitor; N, number; PD-L1, programmed death ligand-1; PS, performance status; * 2 patients received platinum-based chemotherapy + pembrolizumab; ** 1 patient 80%, 3 patients 60%, 2 patient 40%, 1 patient 35%, 1 patient 25%, 1 patient 10%; *** 1 patient 15%.

**Table 2 genes-12-00679-t002:** Key characteristics of patients treated with immune checkpoint blockade.

Characteristic	Total N = 15 (%)
**Treatment**	
Anti-PD1 * Anti-PD-L1 ** Anti-PD-1 + anti-CTLA4 *** Platinum-based chemotherapy + anti-PD1 ^†^	10 (66.7) 2 (13.3) 1 (6.7) 2 (13.3)
**Line of immune checkpoint blockade administration**	
1st ≥2nd	6 (40.0) 9 (60.0)
**PS prior to immune checkpoint blockade**	
0–1 2	11 (73.3) 4 (26.7)

PD-1, programmed death-1; PD-L1, programmed death ligand-1; PS, performance status; * 6 patients pembrolizumab and 4 patients nivolumab; ** 2 patients atezolizumab; *** Nivolumab + ipilimumab; ^†^ 2 patients platinum-pemetrexed + pembrolizumab.

**Table 3 genes-12-00679-t003:** Response to treatment with immune checkpoint blockade.

Best Response	Total N = 15 (%)
Partial response	1 (6.7)
Stable disease	1 (6.7)
Progressive disease	11 (73.3)
Not evaluable	2 * (13.3)

* 2 patients died after only one administration of ICB treatment in the absence of radiological disease reassessment.

**Table 4 genes-12-00679-t004:** Univariate and multivariate analyses of prognostic factors for overall survival.

Factor	Univariate Analysis		Multivariate Analysis	
	HR (95% CI)	*p*-Value	HR (95% CI)	*p*-Value
Age, ≥65 y vs. <65 y	1.09 (0.43–2.81	0.85	-	-
Gender, male vs. female	1.20 (0.50–2.88)	0.67	-	-
Smoking history, never vs. ever	1.16 (0.49–2.75)	0.74	-	-
PS at diagnosis of advanced disease, 2 vs. 0–1	4.79 (1.44–15.82)	**0.01**	8.64 (2.01–35.6)	**0.003**
Lines of treatment, ≤2 vs. >3	2.87 (1.01–8.14)	**0.05**	4.05 (1.30–12.63)	**0.02**
ICB treatment, yes vs. no	2.14 (0.88–5.20)	**0.08**	2.88 (1.12–7.41)	**0.03**
PD-L1, ≥1% vs. <1%	1.69 (0.66–4.32)	0.27	-	-
*TP53*, present vs. absent	1.33 (0.52–3.40)	0.55	-	-

ICB, immune checkpoint blockade; PD-L1, programmed death ligand-1; PS, performance status.

## Data Availability

The datasets generated and analyzed during the current study are available from the corresponding author on reasonable request.

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
