# Peer review of "Sensitivity to Immune Checkpoint Blockade in Advanced Non-Small Cell Lung Cancer Patients with EGFR Exon 20 Insertion Mutations"

_genes, 2021, doi:10.3390/genes12050679_

Round 1

Reviewer 1 Report

In this retrospective study by Metro et al., the authors found that immune checkpoint inhibitors had little efficacy against EGFR ex20 mutant NSCLC. This is an important addition to the literature given the rarity of the mutation. This is best described as a case series and some of the conclusions, especially the inferior OS of patients who received a PD-1 inhibitor is flawed and not entirely supported by the data presented.

  • 30 patients were identified over a 12-year period, during which the treatment of NSCLC had dramatically changed, making it difficult to compare the no IO vs IO outcomes. Effective TKI treatment were not even in trials until recent years but 47% of the non-IO group had received a TKI. Are these patients who inherently do better? There needs to be a much more detailed description of when these ‘control’ patients were treated and what they were treated with before a conclusion of treatment with IO results in worse OS can be made.
  • Please include a discussion of the potential reason of why treatment with IO appears to be detrimental to OS – ie It is not just that patients who receive IO first line do worse, those who received it second line only had an OS of 8.1 months. Is it underpefromance of the IO group (8.1 months OS is not what you would expect with 1st line therapy with a platinum doublet), coupled with overperformance of the non-IO group (25.2 months OS is also not what you would expect without effective TKI)? So are these findings due to chance because of the small numbers? Or is there some other explanation – did the second line IO patients receive standard therapy?
  • Please include in the discussion section the limitations of this study.
  • With such a small cohort, prognostic factors should be prespecified. Multivariate regression is barely possible, and the authors used a backward selection model that ended up including 3 covariates. This is statistically flawed.
  • The numbering of Figure 3 does not match the manuscript text.
  • A bit too many figures and tables – please highlight the important messages and move some to the supplemental.

Author Response

Point by point reply

Reviewer 1 comments

Comment 1: 30 patients were identified over a 12-year period, during which the treatment of NSCLC had dramatically changed, making it difficult to compare the no IO vs IO outcomes. Effective TKI treatment were not even in trials until recent years but 47% of the non-IO group had received a TKI. Are these patients who inherently do better? There needs to be a much more detailed description of when these ‘control’ patients were treated and what they were treated with before a conclusion of treatment with IO results in worse OS can be made.

Reply: we thank the reviewer for this suggestion. In order to make it clearer, we added a supplemental  table (table S1) in which it can be seen that there is some imbalance in terms of treatments with regard to platinum-based chemotherapy which might have influenced the different survival between the IO and the non-IO groups. This has been commented in the discussion (see lines 435-441).

With regard to TKIs, it is true, as suggested by the reviewer, that 47% of patients in the non-IO group received an EGFR-TKI, as opposed to 27% in the IO group, but the TKIs that were administered are generally ineffective for patients with this type of mutation (except for 1 patient receiving poziontinib in the non-IO group), so the difference is survival is more likely due to differences in the receipt of cytotoxic chemotherapy between the 2 groups (as suggested above).

Comment 2: Please include a discussion of the potential reason of why treatment with IO appears to be detrimental to OS – ie It is not just that patients who receive IO first line do worse, those who received it second line only had an OS of 8.1 months. Is it underpefromance of the IO group (8.1 months OS is not what you would expect with 1st line therapy with a platinum doublet), coupled with overperformance of the non-IO group (25.2 months OS is also not what you would expect without effective TKI)? So are these findings due to chance because of the small numbers? Or is there some other explanation – did the second line IO patients receive standard therapy?

Reply: We thank the reviewer for pointing this out. In part, this observation has been answered in the previous point. However, the potential reasons for the difference in OS have been addressed in the discussion (see lines 435-443).

With regard to overperformance of the non-IO group, similar median survivals have been observed by other authors (Naidoo et al. 2015; Xu et al. 2020) and this point has been added in the discussion (see lines 432-435).

Comment 3: Please include in the discussion section the limitations of this study.

Reply: This has been done (see lines 444-448).

Comment 4: With such a small cohort, prognostic factors should be prespecified. Multivariate regression is barely possible, and the authors used a backward selection model that ended up including 3 covariates. This is statistically flawed.

Reply: Thank you for your observation: the use of backward selection methods or more generally lasso selection is widely debated in the medical statistics field, especially in the presence of small samples. In our case, the selection included the 3 covariates that were significant in the univariate approach. Therefore, the specification of the model is the same. We have reformulated the methods section by including this as a model specification criterion (see lines 223-225).

Comment 5: The numbering of Figure 3 does not match the manuscript text

Reply: We have corrected this throughout the text.

Comment 6: A bit too many figures and tables – please highlight the important messages and move some to the supplemental

Reply: We agree with the reviewer and have changed figure 1 (study flowchart) into figure S1

Reviewer 2 Report

In this study, the authors examined the impact of EGFR exon 20 insertion (Ex20ins) mutations on NSCLC response to immune checkpoint blockade (ICB) treatment. It was found that ICB treatment is poorly effective in NSCLCs with EGFR Ex20ins mutations. The author concluded that the information of patients with EGFR Ex20ins mutations being less responsive to ICB treatment is crucial in order to select the optimal treatment strategy for patients with this subtype of EGFR mutation.

Major concerns:

  1. Figure 3 showed that differences in PFS and OS between two study groups that were divided either based on the line of treatment or on the PD-L1 status were not statistically significant. This is likely due to a small sample size, which not only affects the reliability of the results but also generates a bias in the analysis.
  2. Is it possible that the EGFR mutation status is irrelevant to ICB response? To know if patients with EGFR Ex20ins is less responsive to PD-L1 inhibitors than patients with other EGFR mutations, especially the common (L858R+del19) EGFR mutation, the authors should compare the PFS or OS between two lines of treatment (with vs. without PD-L1 inhibitor) and between two different PD-L1 status (less than 1% vs no less than 1%) in patients with common (L858R+del19) EGFR mutation.
  3. Please elaborate how the finding of this study is crucial to the selection of optimal treatment strategy for patients with EGFR Ex20ins. Can the authors say with confidence that patients with EGFR Ex20ins should NOT be given the ICB treatment?

Author Response

Point by point reply

Reviewer 2 comments

Comment 1: Figure 3 showed that differences in PFS and OS between two study groups that were divided either based on the line of treatment or on the PD-L1 status were not statistically significant. This is likely due to a small sample size, which not only affects the reliability of the results but also generates a bias in the analysis.

Reply: We are aware of this limitation although we believe that an analysis on 30 pts with this rare kind of mutation is still worthy of being reported. Therefore, we have acknowledged this problem in the limitations of the study (see lines 444-446).

Comment 2: Is it possible that the EGFR mutation status is irrelevant to ICB response? To know if patients with EGFR Ex20ins is less responsive to PD-L1 inhibitors than patients with other EGFR mutations, especially the common (L858R+del19) EGFR mutation, the authors should compare the PFS or OS between two lines of treatment (with vs. without PD-L1 inhibitor) and between two different PD-L1 status (less than 1% vs no less than 1%) in patients with common (L858R+del19) EGFR mutation.

Reply: We really thank the reviewer for this observation: however, there are many reports published on the (poor) efficacy of ICB in patients with classic EGFR mutations (L858R+del19) and very few (almost none) on ICB in patients with EGFR Ex20ins. For this reason, we decided to focus only on the Ex20ins group of patients that has been often neglected. As a result, the comparison between efficacy of ICB in EGFR Ex20ins versus classic EGFR mutations was beyond the scope of this study.

Comment 3: Please elaborate how the finding of this study is crucial to the selection of optimal treatment strategy for patients with EGFR Ex20ins. Can the authors say with confidence that patients with EGFR Ex20ins should NOT be given the ICB treatment?

Reply: We thank the reviewer for this observation. In the conclusion we did not say that patients with EGFR Ex20ins should not be given the ICB treatment (which cannot be claimed with this study) we just stated that based on our data “immunotherapy should be the least considered option in patients with EGFR Ex20ins mutation, to be reserved as a treatment option only after having administered standard platinum-based chemotherapy” (line 462). However, we agree with the reviewer that data on chemo-immunotherapy in the first line are lacking in these patients, so we added the following phrase: “On the other hand, more efficacy data are needed in order to support the use of ICB in combination with platinum-based chemotherapy as first line treatment for these patients” (line 464).

Round 2

Reviewer 1 Report

no further comments

Reviewer 2 Report

The authors have addressed all my concerns.